# LEARNING ROBUST MODELS USING THE PRINCIPLE OF INDEPENDENT CAUSAL MECHANISMS

## ABSTRACT

Standard supervised learning breaks down under data distribution shift. However, the principle of independent causal mechanisms (ICM, Peters et al. (2017)) can turn this weakness into an opportunity: one can take advantage of distribution shift between different environments during training in order to obtain more robust models. We propose a new gradient-based learning framework whose objective function is derived from the ICM principle. We show theoretically and experimentally that neural networks trained in this framework focus on relations remaining invariant across environments and ignore unstable ones. Moreover, we prove that the recovered stable relations correspond to the true causal mechanisms under certain conditions. In both regression and classification, the resulting models generalize well to unseen scenarios where traditionally trained models fail.

## 1 INTRODUCTION

Standard supervised learning has shown impressive results when training and test samples follow the same distribution. However, many real world applications do not conform to this setting, so that research successes do not readily translate into practice (Lake et al., 2017). The task of *Domain Generalization* (DG) addresses this problem: it aims at training models that generalize well under domain shift. In contrast to domain *adaption*, where a few labeled and/or many unlabeled examples are provided for each target test domain, in DG absolutely no data is available from the test domains' distributions making the problem unsolvable in general.

In this work, we view the problem of DG specifically using ideas from causal discovery. To make the problem of DG well-posed from this viewpoint, we assume that there exists a feature vector $h^\star(\mathbf{X})$ whose relation to the target variable $Y$ is invariant across all environments. Consequently, the conditional probability $p(Y \mid h^\star(\mathbf{X}))$ has predictive power in each environment. From a causal perspective, changes between domains or environments can be described as interventions; and causal relationships – unlike purely statistical ones – remain invariant across environments unless explicitly changed under intervention. This is due to the fundamental principle of "Independent Causal Mechanisms" which will be discussed in Section 3. From a causal standpoint, finding robust models is therefore a *causal discovery* task (Bareinboim & Pearl, 2016; Meinshausen, 2018). Taking a causal perspective on DG, we aim at identifying features which (i) have an invariant relationship to the target variable $Y$ and (ii) are maximally informative about $Y$.

This problem has already been addressed with some simplifying assumptions and a discrete combinatorial search in Magliacane et al. (2018); Rojas-Carulla et al. (2018), but we make weaker assumptions and use gradient based optimization. Gradient based optimization is attractive because it readily scales to high dimensions and offers the possibility to *learn* very informative features, instead of merely selecting among predefined ones. Approaches to invariant relations similar to ours were taken in Ghassami et al. (2017), who restrict themselves to linear relations, and Arjovsky et al. (2019); Krueger et al. (2020), who minimize an invariant empirical risk objective.

Problems (i) and (ii) are quite intricate because the search space has combinatorial complexity and testing for conditional independence in high dimensions is notoriously difficult. Our main contributions to this problem are the following:

- By connecting invariant (causal) relations with normalizing flows, we propose a differentiable two-part objective of the form $I(Y; h(\mathbf{X})) + \lambda_I \mathcal{L}_I$, where $I$ is the mutual information

and $\mathcal{L}_I$ enforces the invariance of the relation between $h(\mathbf{X})$ and $Y$ across all environments. This objective operationalizes the ICM principle with a trade-off between feature informativeness and invariance controlled by parameter $\lambda_I$. Our formulation generalizes existing work because our objective is not restricted to linear models.

- We take advantage of the continuous objective in three important ways: (1) We can learn invariant new features, whereas graph-based methods can only select features from a predefined set. (2) Our approach does not suffer from the scalability problems of combinatorial optimization methods as proposed in e.g. Peters et al. (2016) and Rojas-Carulla et al. (2018). (3) Our optimization via normalizing flows, i.e. in the form of a density estimation task, facilitates accurate maximization of the mutual information.

- We show how our objective simplifies in important special cases and under which conditions its optimal solution identifies the true causal parents of the target variable $Y$. We empirically demonstrate that the new method achieves good results on two datasets proposed in the literature.

## 2 RELATED WORK

Different types of invariances have been considered in the field of DG. One type is defined on the feature level, i.e. features $h(\mathbf{X})$ are invariant across environments if they follow the same distribution in all environments (e.g. Pan et al. (2010); Ganin et al. (2016); Ben-David et al. (2007)). However, this form of invariance is problematic since for instance the distribution of the target variable might change between environments. In this case we might expect that the distribution $h(\mathbf{X})$ changes as well. A more plausible and theoretically justified type of invariance is the invariance of relations (Peters et al., 2016; Magliacane et al., 2018; Rojas-Carulla et al., 2018). A relation between a target $Y$ and some features is invariant across environments, if the conditional distribution of $Y$ given the features is the same for all environments. Existing approaches model a conditional distribution for each feature selection and check for the invariance property (Peters et al., 2016; Rojas-Carulla et al., 2018; Magliacane et al., 2018). However, this does not scale well. We provide a theoretical result connecting *normalizing flows* and *invariant relations* which in turn allows for gradient-based learning of the problem. In order to exploit our formulation, we also use the Hilbert-Schmidt-Independence Criterion that has been used for robust learning by Greenfeld & Shalit (2019) in the one environment setting. Arjovsky et al. (2019) propose a gradient-based learning framework which exploits a weaker notion of invariance. Their definition is only a necessary condition, but does not guarantee the more causal definition of invariance we treat in this work. The connection between DG, invariances and causality has been pointed out for instance by Meinshausen (2018); Rojas-Carulla et al. (2018); Zhang et al. (2015). From a causal perspective, DG is a causal discovery task (Meinshausen, 2018).

For studies on causal discovery in the purely observational setting see e.g. Spirtes & Glymour (1991); Pearl (2009); Chickering (2002), but they cannot take advantage of variations across environments. The case of different environments has been studied by Hoover (1990); Tian & Pearl (2001); Mooij et al. (2016); Peters et al. (2016); Bareinboim & Pearl (2016); Magliacane et al. (2018); Ghassami et al. (2018); Huang et al. (2020). Most of these approaches rely on combinatorial methods based on graphical models or are restricted to linear mechanisms, whereas our model defines a continuous objective for very general non-linear models. The distinctive property of causal relations to remain invariant across environments in the absence of direct interventions has been known since at least the 1930s (Frisch, 1938; Heckman & Pinto, 2013). However, its crucial role as a tool for causal discovery was – to the best of our knowledge– only recently recognized by Peters et al. (2016). Their estimator – *Invariant Causal Prediction* (ICP) – returns the intersection of all subsets of variables that have an invariant relation w.r.t. $Y$. The output is shown to be the set of the direct causes of $Y$ under suitable conditions. However, their method assumes an underlying linear model and must perform an exhaustive search over all possible variable sets $\mathbf{X}_T$, which does not scale. Extensions to time series and non-linear additive noise models were studied in Heinze-Deml et al. (2018); Pfister et al. (2019). Our treatment of invariance is inspired by these papers and also discusses identifiability results, i.e. conditions when the identified variables are indeed the direct causes. Key differences between ICP and our approach are the following: Firstly, we propose a formulation that allows for a gradient-based learning without strong assumptions on the underlying causal model such as linearity. Second, while ICP tends to exclude features from the parent set when in doubt, our algorithm prefers to err in the direction of best prediction performance in this case.

# 3 PRELIMINARIES

In the following we introduce the basics of this article as well as the connection between DG and causality. Basics on causality are presented in Appendix A. We first define our notation as follows: We denote the set of all variables describing the system under study as $\widetilde{\mathbf{X}} = \{X_1, \ldots, X_D\}$. One of these variables will be singled out as our prediction target, whereas the remaining ones are observed and may serve as predictors. To clarify notation, we call the target variable $Y \equiv X_i$ for some $i \in \{1, \ldots, D\}$, and the remaining observations are $\mathbf{X} = \widetilde{\mathbf{X}} \backslash \{Y\}$. Realizations of a random variable are denoted with lower case letters, e.g. $x_i$. We assume that observations can be obtained in different environments $e \in \mathcal{E}$. Symbols with superscript, e.g. $Y^e$, refer to a specific environment, whereas symbols without refer to data pooled over all environments. We distinguish known environments $e \in \mathcal{E}_{\text{seen}}$, where training data are available, from unknown ones $e \in \mathcal{E}_{\text{unseen}}$, where we wish our models to generalize to. The set of all environments is $\mathcal{E} = \mathcal{E}_{\text{seen}} \cup \mathcal{E}_{\text{unseen}}$. We assume that all random variables have a density $p_A$ with probability distribution $P_A$ (for some variable or set $A$). We consider the environment to be a random variable $E$ and therefore a system variable similar to Mooij et al. (2016). This gives an additional view on casual discovery and the DG problem.

Independence and dependence of two variables $A$ and $B$ is written as $A \perp B$ and $A \not\perp B$ respectively. Two random variables $A, B$ are conditionally independent given $C$ if $P(A, B \mid C) = P(A \mid C)p(B \mid C)$. This is denoted with $A \perp B \mid C$. Intuitively, it means $A$ does not contain any information about $B$ if $C$ is known (for details see e.g. Peters et al., 2017). Similarly, one can define independence and conditional independence for sets of random variables.

## 3.1 INVARIANCE AND THE PRINCIPLE OF ICM

DG is in general unsolvable because distributions between seen and unseen environments could differ arbitrarily. In order to transfer knowledge from $\mathcal{E}_{\text{seen}}$ to $\mathcal{E}_{\text{unseen}}$, we have to make assumptions on how seen and unseen environments relate. These assumptions have a close link to causality.

We assume certain relations between variables remain invariant across all environments. A subset $\mathbf{X}_S \subset \mathbf{X}$ of variables *elicits an invariant relation* or *satisfies the invariance property* w.r.t. $Y$ over a subset $W \subset \mathcal{E}$ of environments if

$$\forall e, e' \in W: \quad P(Y^e \mid \mathbf{X}_S^e = u) = P(Y^{e'} \mid \mathbf{X}_S^{e'} = u) \tag{1}$$

for all $u$ where both conditional distributions are well-defined. Equivalently, we can define the invariance property by $Y \perp E \mid \mathbf{X}_S$ and $I(Y; E \mid \mathbf{X}_S) = 0$ for $E$ restricted to $W$. The *invariance property* for computed features $h(\mathbf{X})$ is defined analogously by the relation $Y \perp E \mid h(\mathbf{X})$.

Although we can only test for (equation 1) in $\mathcal{E}_{\text{seen}}$, taking a causal perspective allows us to derive plausible conditions – expressed by Assumptions 1 and 2 – for an invariance to remain valid in all environments $\mathcal{E}$. In brief, we assume that environments correspond to interventions in the system and invariance arises from the principle of *independent causal mechanisms* (Peters et al., 2017, ICM).

At first, consider the joint density $p_{\widetilde{\mathbf{X}}}(\widetilde{\mathbf{X}})$. The chain rule offers a combinatorial number of ways to decompose this distribution into a product of conditionals. Among those, the *causal factorization*

$$p_{\widetilde{\mathbf{X}}}(x_1, \ldots, x_D) = \prod_{i=1}^{D} p_i(x_i \mid \mathbf{x}_{pa(i)}) \tag{2}$$

is singled out by conditioning each $X_i$ onto its *causal parents* or *direct causes* $\mathbf{X}_{pa(i)}$, where $pa(i)$ denotes the appropriate index set. The special properties of this factorization are discussed in Peters et al. (2017). The conditionals $p_i$ of the causal factorization are called *causal mechanisms*. An *intervention* onto the system is defined by replacing one or several factors in the decomposition with different (conditional) densities $\bar{p}$. Here, we distinguish *soft-interventions* where $\bar{p}_j(x_j \mid \mathbf{x}_{pa(j)}) \neq p_j(x_j \mid \mathbf{x}_{pa(j)})$ and *hard-interventions* where $\bar{p}_j(x_j \mid \mathbf{x}_{pa(j)}) = \bar{p}_j(x_j)$ is a density which does not depend on $x_{pa(j)}$ (e.g. an atomic intervention where $x_j$ is forced to take a specific value $\bar{x}$). The resulting joint distribution for a single intervention is

$$\bar{p}_{\widetilde{\mathbf{X}}}(x_1, \ldots, x_D) = \bar{p}_j(x_j \mid \mathbf{x}_{pa(j)}) \prod_{i=1, i \neq j}^{D} p_i(x_i \mid \mathbf{x}_{pa(i)}) \tag{3}$$

and extends to multiple simultaneous interventions in the obvious way. The principle of *independent causal mechanisms* (ICM) states that every mechanism acts independently of the others (Peters et al., 2017). Consequently, an intervention replacing $p_j$ with $\bar{p}_j$ has no effect on the other factors $p_{i \neq j}$, as indicated by equation 3. This is a crucial property of the causal decomposition – alternative factorizations do not exhibit this behavior. Instead, a coordinated modification of several factors is generally required to model the effect of an intervention in a non-causal decomposition.

We utilize this principle as a tool to train *robust* models. To do so, we make two additional assumptions, similar to Peters et al. (2016) and Heinze-Deml et al. (2018):

**Assumption 1.** Any differences in the joint distributions $p_{\mathbf{X}}^e$ from one environment to the other are fully explainable as interventions: replacing factors $p_i^e(x_i \mid \mathbf{x}_{pa(i)})$ in environment $e$ with factors $p_i^{e'}(x_i \mid \mathbf{x}_{pa(i)})$ in environment $e'$ (for some subset of the variables) is the only admissible change.

**Assumption 2.** The mechanism $p(y \mid \mathbf{x}_{pa(Y)})$ for the target variable is invariant under changes of environment. In other words, we require conditional independence $Y \perp E \mid \mathbf{X}_{pa(Y)}$.

Assumption 2 implies that $Y$ must not directly depend on $E$. In addition, it has important consequences when there exist omitted variables $H$, which influence $Y$ but have not been measured. Specifically, if the omitted variables depend on the environment (hence $H \not\perp E$) or $H$ contains a hidden confounder of $\mathbf{X}_{pa(Y)}$ and $Y$ (the system is not causally sufficient and $\mathbf{X}_{pa(Y)}$ becomes a "collider", hence $H \not\perp E \mid \mathbf{X}_{pa(Y)}$), then $Y$ and $E$ are no longer $d$-separated by $\mathbf{X}_{pa(Y)}$ and Assumption 2 is unsatisfiable. Then our method will be unable to find an invariant mechanism (see Appendix B for more details).

If we knew the causal decomposition, we could use these assumptions directly to train a robust model for $Y$ – we would simply regress $Y$ on its parents $\mathbf{X}_{pa(Y)}$. However, we only require that a causal decomposition with these properties exists, but do not assume that it is known. Instead, our method uses the assumptions indirectly – by simultaneously considering data from different environments – to identify a stable regressor for $Y$.

We call a regressor stable if it solely relies on predictors whose relationship to $Y$ remains invariant across environments, i.e. is not influenced by any intervention. By assumption 2, such a regressor always exists. However, predictor variables beyond $\mathbf{X}_{pa(Y)}$ may be used as well, e.g. children of $Y$ or parents of children, provided their relationships to $Y$ do not depend on the environment. The case of children is especially interesting: Suppose $X_j$ is a noisy measurement of $Y$, described by the causal mechanism $P_j(X_j \mid Y)$. As long as the measurement device works identically in all environments, including $X_j$ as a predictor of $Y$ is desirable, despite it being a child. We discuss and illustrate Assumption 2 in Appendix B. In general, prediction accuracy will be maximized when all suitable predictor variables are included into the model. Accordingly, our algorithm will asymptotically identify the full set of stable predictors for $Y$. In addition, we will prove under which conditions this set contains exactly the parents of $Y$. Note that there are different ideas on whether most supervised learning tasks conform to this setting (Schölkopf et al., 2012; Arjovsky et al., 2019).

## 3.2    DOMAIN GENERALIZATION

In order to exploit the principle of ICM for DG, we formulate the DG problem as follows

$$h^\star := \underset{h \in \mathcal{H}}{\arg\max} \left\{ \min_{e \in \mathcal{E}} I(Y^e; h(\mathbf{X}^e)) \right\} \quad \text{s.t.} \quad Y \perp E \mid h(\mathbf{X}) \tag{4}$$

where $h \in \mathcal{H}$ denotes a learnable feature extraction function $h \colon \mathbb{R}^D \to \mathbb{R}^M$ where $M$ is a hyperparameter. This optimization problem defines a maximin objective: The features $h(\mathbf{X})$ should be as informative as possible about the response $Y$ even in the most difficult environment, while conforming to the ICM constraint that the relationship between features and response must remain invariant across all environments. In principle, our approach can also optimize related objectives like the average mutual information over environments. However, very good performance in a majority of the environments could then mask failure in a single (outlier) environment. We opted for the maximin formulation to avoid this.

As it stands, equation 4 is hard to optimize, because traditional independence tests for the constraint $Y \perp E \mid h(\mathbf{X})$ cannot cope with conditioning variables selected from an infinitely large space $\mathcal{H}$. A re-formulation of the DG problem to solve this problem is our main theoretical contribution.

### 3.3 Normalizing Flows

Normalizing flows form a class of probabilistic models that has recently received considerable attention, see Papamakarios et al. (2019) for an in-depth review. They model complex distributions by means of invertible functions $T$ (chosen from some model space $\mathcal{T}$) which map the densities of interest to latent normal distributions. The inverses $F = T^{-1}$ then act as generative models for the target distributions. Normalizing flows are typically built with specialized neural networks that are invertible by construction and have tractable Jacobian determinants.

In our case, we represent the conditional distribution $P(Y \mid h(\mathbf{X}))$ using a *conditional* normalizing flow. To this end, we seek a mapping $R = T(Y; h(\mathbf{X}))$ that is diffeomorphic in $Y$ such that $R \sim \mathcal{N}(0, 1) \perp h(\mathbf{X})$ when $Y \sim P(Y \mid h(\mathbf{X}))$. This is a generalization of the well-studied additive Gaussian noise model $R = Y - f(h(\mathbf{X}))$, see section 4.2. The inverse $Y = F(R; h(\mathbf{X}))$ assumes the role of a structural equation for the mechanism $p(Y \mid h(\mathbf{X}))$ with $R$ being the corresponding noise variable.[1] However, in our context it is most natural to learn $T$ (rather than $F$) by minimizing the negative log-likelihood (NLL) of $Y$ under $T$ (Papamakarios et al., 2019), which takes the form

$$\mathcal{L}_{\text{NLL}}(T, h) := \mathbb{E}_{h(\mathbf{X}), Y}\left[\|T(Y; h(\mathbf{X})\|^2/2 - \log|\det \nabla_y T(Y; h(\mathbf{X}))|\right] + C \qquad (5)$$

where $\det \nabla_y T$ is the Jacobian determinant and $C = \dim(Y)\log(\sqrt{2\pi})$ is a constant that can be dropped. If we consider the NLL on a particular environment $e \in \mathcal{E}$, we denote this with $\mathcal{L}_{\text{NLL}}^e$. Lemma 1 shows that normalizing flows optimized by NLL are indeed applicable to our problem:

**Lemma 1.** *(proof in Appendix C) Let $h^\star, T^\star := \arg\min_{h \in \mathcal{H}, T \in \mathcal{T}} \mathcal{L}_{\text{NLL}}(T, h)$ be the solution of the NLL minimization problem on a sufficiently rich function space $\mathcal{T}$. Then the following properties are guaranteed for arbitrary sets $\mathcal{H}$ of feature extractors:*

  *(a) $h^\star$ also maximizes the mutual information, i.e. $h^\star = g^\star$ with $g^\star = \arg\max_{g \in \mathcal{H}} I(g(\mathbf{X}); Y)$*

  *(b) $h^\star$ is independent of the flow's latent variable: $h^\star(\mathbf{X}) \perp R$ with $R = T^\star(Y; h^\star(\mathbf{X}))$.*

Statement (a) guarantees that $h^\star$ extracts as much information about $Y$ as possible. Hence, the objective (4) becomes equivalent to optimizing (5) when we restrict the space $\mathcal{H}$ of admissible feature extractors to the subspace $\mathcal{H}_\perp$ satisfying the invariance constraint $Y \perp E \mid h(\mathbf{X})$: $\arg\min_{h \in \mathcal{H}_\perp} \max_{e \in \mathcal{E}} \min_{T \in \mathcal{T}} \mathcal{L}_{\text{NLL}}^e(T; h) = \arg\max_{h \in \mathcal{H}_\perp} \min_{e \in \mathcal{E}} I(Y^e; h(\mathbf{X}^e))$ (Appendix C). Statement (b) ensures that the flow indeed implements a valid structural equation, which requires that $R$ can be sampled independently of the features $h(\mathbf{X})$. More details about normalizing flows can be found in Appendix C.

## 4 Method

In the following we propose a way of indirectly expressing the constraint in equation 4 via normalizing flows. Thereafter, we combine this result with Lemma 1 to obtain a differentiable objective for solving the DG problem. We also present important simplifications for least squares regression and softmax classification and discuss relations of our approach with causal discovery.

### 4.1 Learning the Invariance Property

The following theorem establishes a connection between invariant relations, prediction residuals and normalizing flows. The key consequence is that a suitably trained normalizing flow translates the statistical independence of the latent variable $R$ from the features and environment $(h(\mathbf{X}), E)$ into the desired invariance of the mechanism $P(Y \mid h(\mathbf{X}))$ under changes of $E$. We will exploit this for an elegant reformulation of the DG problem (4) into the objective (7) below.

**Theorem 1.** *Let $h$ be a differentiable function and $Y, \mathbf{X}, E$ be random variables. Furthermore, let $R = T(Y; h(\mathbf{X}))$ be a continuous, differentiable function that is a diffeomorphism in $Y$. Suppose that $R \perp (h(\mathbf{X}), E)$. Then, it holds that $Y \perp E \mid h(\mathbf{X})$.*

*Proof.* The decomposition rule for the assumption $R \perp (h(\mathbf{X}), E)$ (1) implies $R \perp h(\mathbf{X})$ (2). To simplify notation, we define $Z := h(\mathbf{X})$. Because $T$ is invertible in $Y$ and due to the change of

---

[1] $F$ is the concatenation of the normal CDF with the inverse CDF of $P(Y \mid h(\mathbf{X}))$, see Peters et al. (2014).

variables (c.o.v.) formula, we obtain

$$p_{Y|Z,E}(y \mid z, e) \stackrel{(c.o.v.)}{=} p_{R|Z,E}(T(y,z) \mid z, e) \left| \det \frac{\partial T}{\partial y}(y,z) \right| \stackrel{(1)}{=} p_R(r) \left| \det \frac{\partial T}{\partial y}(y,z) \right|$$

$$\stackrel{(2)}{=} p_{R|Z}(r \mid z) \left| \det \frac{\partial T}{\partial y}(y,z) \right| \stackrel{(c.o.v.)}{=} p_{Y|Z}(y \mid z).$$

This implies $Y \perp E \mid Z$. $\qquad\square$

The theorem states in particular that if there exists a suitable diffeomorphism $T$ such that $R \perp (h(\mathbf{X}), E)$, then $h(\mathbf{X})$ satisfies the invariance property w.r.t. $Y$. Note that if Assumption 2 is violated, the condition $R \perp (h(\mathbf{X}), E)$ is unachievable in general and therefore the theorem is not applicable (see Appendix B). We use Theorem 1 in order to *learn* features $h$ that meet this requirement. In the following, we denote a conditional normalizing flow parameterized via $\theta$ with $T_\theta$. Furthermore, $h_\phi$ denotes a feature extractor implemented as a neural network parameterized via $\phi$. We can relax condition $R \perp (h_\phi(\mathbf{X}), E)$ by means of the Hilbert Schmidt Independence Criterion (HSIC), a kernel-based independence measure (see Appendix D for the mathematical definition and Gretton et al. (2005) for details). This loss, denoted as $\mathcal{L}_I$, penalizes dependence between the distributions of $R$ and $(h_\phi(\mathbf{X}), E)$. The HSIC guarantees that

$$\mathcal{L}_I(P_R, P_{h_\phi(\mathbf{X}),E}) = 0 \quad \Longleftrightarrow \quad R \perp (h_\phi(\mathbf{X}), E) \tag{6}$$

where $R = T_\theta(Y; h_\phi(\mathbf{X}))$ and $P_R, P_{h_\phi(\mathbf{X}),E}$ are the distributions implied by the parameter choices $\phi$ and $\theta$. Due to Theorem 1, minimization of $\mathcal{L}_I(P_R, P_{h_\phi(\mathbf{X}),E})$ w.r.t. $\phi$ and $\theta$ will thus approximate the desired invariance property $Y \perp E \mid h_\phi(\mathbf{X})$, with exact validity upon perfect convergence.

When $R \perp (h_\phi(\mathbf{X}), E)$ is fulfilled, the decomposition rule implies $R \perp E$ as well. However, if the differences between environments are small, empirical convergence is accelerated by adding a Wasserstein loss which explicitly enforces the latter, see Appendix D and section 5.2 for details.

## 4.2 EXPLOITING INVARIANCES FOR PREDICTION

Equation 4 can be re-formulated as a differentiable loss using a Lagrange multiplier $\lambda_I$ on the HSIC loss. $\lambda_I$ acts as a hyperparameter to adjust the trade-off between the invariance property of $h_\phi(\mathbf{X})$ w.r.t. $Y$ and the mutual information between $h_\phi(\mathbf{X})$ and $Y$. See Appendix E for algorithm details.

**Normalizing Flows** Using Lemma 1(a), we maximize $\min_{e \in \mathcal{E}} I(Y^e; h_\phi(\mathbf{X}^e))$ by minimizing $\max_{e \in \mathcal{E}} \{\mathcal{L}_{\text{NLL}}(T_\theta, h_\phi)\}$ w.r.t. $\phi, \theta$. To achieve the described trade-off between goodness-of-fit and invariance, we therefore optimize

$$\arg\min_{\theta, \phi} \left( \max_{e \in \mathcal{E}} \left\{ \mathcal{L}_{\text{NLL}}(T_\theta, h_\phi) \right\} + \lambda_I \mathcal{L}_I(P_R, P_{h_\phi(\mathbf{X}),E}) \right) \tag{7}$$

where $R^e = T_\theta(Y^e, h_\phi(\mathbf{X}^e))$ and $\lambda_I > 0$. The first term maximizes the mutual information between $h_\phi(\mathbf{X})$ and $Y$ in the environment where the features are least informative about $Y$ and the second term aims to ensure an invariant relation.

**Additive Noise** Let $f_\theta$ be a regression function. Solving for the noise term gives $R = Y - f_\theta(\mathbf{X})$ which corresponds to a diffeomorphism in $Y$, namely $T_\theta(Y; X) = Y - f_\theta(\mathbf{X})$. If we make two simplified assumptions: (i) the noise is gaussian with zero mean and (ii) $R \perp f_\phi(\mathbf{X})$, then we obtain

$$I(Y; f_\theta(\mathbf{X})) = H(Y) - H(Y \mid f_\theta(\mathbf{X})) = H(Y) - H(R \mid f_\theta(\mathbf{X}))$$

$$\stackrel{(ii)}{=} H(Y) - H(R) \stackrel{(i)}{=} H(Y) - 1/2 \log(2\pi e \sigma^2)$$

where $\sigma^2 = \mathbb{E}[(Y - f_\theta(\mathbf{X}))^2]$. In this case maximizing the mutual information $I(Y; f_\theta(\mathbf{X}))$ amounts to minimizing $\mathbb{E}[(Y - f_\theta(\mathbf{X}))^2]$ w.r.t. $\theta$, i.e. the standard L2-loss. From this, we obtain a simplified version of equation 4 via

$$\arg\min_{\theta} \left( \max_{e \in \mathcal{E}_{\text{seen}}} \left\{ \mathbb{E}[(Y^e - f_\theta(\mathbf{X}^e))^2] \right\} + \lambda_I \mathcal{L}_I(P_R, P_{f_\theta(\mathbf{X}),E}) \right) \tag{8}$$

where $R^e = Y - f_\theta(\mathbf{X}^e)$ and $\lambda_I > 0$. Under the conditions stated above, the objective achieves the mentioned trade-off between information and invariance.

Alternatively we can view the problem as to find features $h_\phi \colon \mathbb{R}^D \to \mathbb{R}^m$ such that $I(h_\phi(\mathbf{X}), Y)$ gets maximized under the assumption that there exists a model $f_\theta(h_\phi(\mathbf{X})) + R = Y$ where $R$ is independent of $h(\mathbf{X})$ and $R$ is gaussian. In this case we obtain similarly as above the learning objective

$$\arg\min_{\theta, \phi} \Big( \max_{e \in \mathcal{E}_{\text{seen}}} \Big\{ \mathbb{E}\big[(Y^e - f_\theta(h_\phi(\mathbf{X}^e)))^2\big] \Big\} + \lambda_I \mathcal{L}_I(P_R, P_{h_\phi(\mathbf{X}), E}) \Big) \tag{9}$$

**Classification**   The expected cross-entropy loss is given through

$$-\mathbb{E}_{\mathbf{X}, Y}\Big[ f(\mathbf{X})_Y - \log\Big( \sum_c \exp(f(\mathbf{X})_c) \Big) \Big]$$

where $f \colon \mathcal{X} \to \mathbb{R}^m$ returns the logits. Minimizing the expected cross-entropy loss amounts to maximizing the mutual information between $f(\mathbf{X})$ and $Y$ (Qin & Kim, 2019; Barber & Agakov, 2003, eq. 3). Let $T(Y; f(\mathbf{X})) = Y \cdot \text{softmax}(f(\mathbf{X}))$ with component-wise multiplication, then $T$ is invertible in $Y$ conditioned on the softmax output. Now we can apply the same invariance loss as above in order to obtain a solution to equation 4.

### 4.3   RELATION TO CAUSAL DISCOVERY

Under certain conditions, solving equation 4 leads to features which correspond to the direct causes of $Y$ (identifiability). In this case, we obtain the causal mechanism by computing the conditional distribution of $Y$ given the direct causes. Therefore equation 4 can also be seen as approximation of the causal mechanism when the identifiability conditions are met. The following Proposition states under which assumptions the direct causes of $Y$ can be recovered by exploiting Theorem 1.

**Proposition 1.** *We assume that the underlying causal graph $G$ is faithful with respect to $P_{\widetilde{\mathbf{X}}, E}$. We further assume that every child of $Y$ in $G$ is also a child of $E$ in $G$. A variable selection $h(\mathbf{X}) = \mathbf{X}_T$ corresponds to the direct causes if the following conditions are met: (i) $T(Y; (X)) \perp E, h(\mathbf{X})$ is satisfied for a diffeomorphism $T(\cdot; h(\mathbf{X}))$, (ii) $h(\mathbf{X})$ is maximally informative about $Y$ and (iii) $h(\mathbf{X})$ contains only variables from the Markov blanket of $Y$.*

The Markov blanket of $Y$ is the only set of vertices which are necessary to predict $Y$ (see Appendix A). We give a proof of Proposition 1 as well as a discussion in Appendix F.

For reasons of explainability and for the task of causal discovery, we employ a gating function $h_\phi$ in order to obtain a variable selection. We use the same gating function as in Kalainathan et al. (2018). The gating function $h_\phi$ represents a 0-1 mask of the input. A complexity loss $\mathcal{L}(\phi)$ represents how many variables are selected and therefore penalizes to include variables. Intuitively speaking, if we search for a variable selection that conforms to the conditions in Proposition 1, the complexity loss would exclude all non-task relevant variables. Therefore, if $\mathcal{H}$ is the set of gating functions, then $h^\star$ in equation 4 would correspond to the direct causes of $Y$ under the conditions listed in Proposition 1. The complexity loss as well as the gating function can be optimized by gradient descent.

## 5   EXPERIMENTS

### 5.1   SYNTHETIC CAUSAL GRAPHS

To evaluate our methods for the regression case, we follow the experimental design of Heinze-Deml et al. (2018). It rests on the causal graph in Figure 1. Each variable $X_1, ..., X_6$ is chosen as the regression target $Y$ in turn, so that a rich variety of local configurations around $Y$ is tested. The corresponding structural equations are selected among four model types of the form $f(\mathbf{X}_{pa(i)}, N_i) = \sum_{j \in pa(i)} \text{mech}(a_j X_j) + N_i$, where 'mech' is either the identity (hence we get a linear SCM), Tanhshrink, Softplus or ReLU, and one multiplicative noise mechanism of the form $f_i(\mathbf{X}_{pa(i)}, N_i) = (\sum_{j \in pa(i)} a_j X_j) \cdot (1 + (1/4)N_i) + N_i$, resulting in 1365 different settings. For each setting, we define an observational environment (using exactly the selected mechanisms) and three interventional ones, where soft or do-interventions are applied to non-target variables according to Assumptions 1 and 2 (full details in Appendix G). Each inference model is trained on 1024 realizations of three environments, whereas the fourth one is held back for DG testing. The tasks are to identify the parents of the current target variable $Y$, and to train a transferable regression model based on this parent hypothesis. We measure performance by the accuracy of the detected parent sets and by the L2 regression errors relative to the regression function using the ground-truth parents.

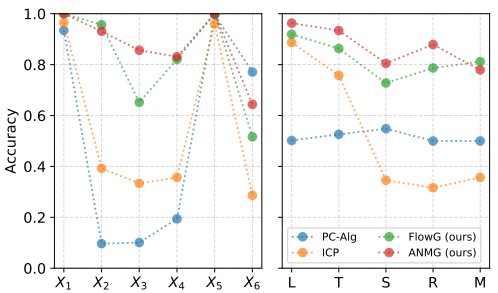 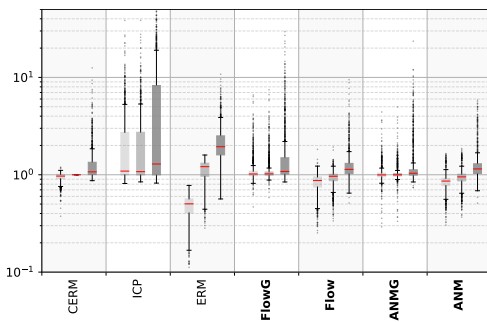

Figure 2: Detection accuracy of the direct causes for baselines and our gating architectures, broken down for different target variables (left) and mechanisms (right: **L**inear, **T**anhshrink, **S**oftplus, **R**eLU, **M**ultipl. Noise).

Figure 3: Logarithmic plot of L2 errors, normalized by CERM test error. For each method (ours in bold) from left to right: training error, test error on seen environments, domain generalization error on unseen environments.

We evaluate four models derived from our theory: two normalizing flows as in equation 4 with and without gating mechanisms (FlowG, Flow) and two additive noise models, again with and without gating mechanism (ANMG, ANM), using a feed-forward network with the objective in equation 9 (ANMG) and equation 8 (ANM). For comparison, we train three baselines: ICP (a causal discovery algorithm also exploiting ICM, but restricted to linear regression, Peters et al. (2016)), a variant of the PC-Algorithm (PC-Alg, see Appendix G.4) and standard empirical-risk-minimization ERM, a feed-forward network minimizing the L2-loss, which ignores the causal structure by regressing $Y$ on all other variables. We normalize our results with a ground truth model (CERM), which is identical to ERM, but restricted to the true causal parents of the respective $Y$.

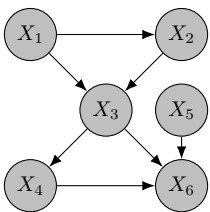

Figure 1: Directed graph of our SCM. Target variable $Y$ is chosen among $X_1, \ldots, X_6$ in turn.

The accuracy of parent detection is shown in Figure 2. The score indicates the fraction of the experiments where the exact set of all causal parents was found and all non-parents were excluded. We see that the PC algorithm performs unsatisfactorily, whereas ICP exhibits the expected behavior: it works well for variables without parents and for linear SCMs, i.e. exactly within its specification. Among our models, only the gating ones explicitly identify the parents. They clearly outperform the baselines, with a slight edge for ANMG, as long as its assumption of additive noise is fulfilled.

Figure 3 and Table 1 report regression errors for seen and unseen environments, with CERM indicating the theoretical lower bound. The PC algorithm is excluded from this experiment due to its poor detection of the direct causes. ICP wins for linear SCMs, but otherwise has largest errors, since it cannot accurately account for non-linear mechanisms. ERM gives reasonable test errors (it even overfits the training data), but generalizes poorly to unseen environments, as expected. Our models perform quite similarly to CERM. We again find a slight edge for ANMG, except under multiplicative noise, where ANMG's additive noise assumption is violated and Flow is superior. All methods (including CERM) occasionally fail in the domain generalization task, indicating that some DG problems are more difficult than others, e.g. when the differences between seen environments are too small to reliably identify the invariant mechanism or the unseen environment requires extrapolation beyond the training data boundaries. Models without gating (Flow, ANM) seem to be slightly more robust in this respect. A detailed analysis of our experiments can be found in Appendix G.

## 5.2 COLORED MNIST

To demonstrate that our model is able to perform DG in the classification case, we use the same data generating process as in the colored variant of the MNIST-dataset established by Arjovsky et al. (2019), but create training instances online rather than upfront. The response is reduced to two labels – 0 for all images with digit $\{0, \ldots, 4\}$ and 1 for digits $\{5, \ldots 9\}$ – with deliberate label noise

| Models | Linear | Tanhshrink | Softplus | ReLU | Mult. Noise |
|---|---|---|---|---|---|
| FlowG (ours) | $1.05..._{4.2}$ | $1.08..._{4.8}$ | $1.09..._{5.52}$ | $1.08..._{5.7}$ | $1.55..._{8.64}$ |
| ANMG (ours) | $1.02..._{1.56}$ | $\mathbf{1.03}..._{2.23}$ | $\mathbf{1.04}..._{4.66}$ | $\mathbf{1.03}..._{4.32}$ | $1.46..._{4.22}$ |
| Flow (ours) | $1.08..._{1.61}$ | $1.14..._{1.57}$ | $1.14..._{1.55}$ | $1.14..._{1.54}$ | $\mathbf{1.35}..._{4.07}$ |
| ANM (ours) | $1.05..._{1.52}$ | $1.15..._{1.47}$ | $1.14..._{1.47}$ | $1.15..._{1.54}$ | $1.48..._{4.19}$ |
| ICP (Peters et al., 2016) | $\mathbf{0.99}..._{25.7}$ | $1.44..._{20.39}$ | $3.9..._{23.77}$ | $4.37..._{23.49}$ | $8.94..._{33.49}$ |
| ERM | $1.79..._{3.84}$ | $1.89..._{3.89}$ | $1.99..._{3.71}$ | $2.01..._{3.62}$ | $2.08..._{5.86}$ |
| CERM (true parents) | $1.06..._{1.89}$ | $1.06..._{1.84}$ | $1.06..._{2.11}$ | $1.07..._{2.15}$ | $1.37..._{5.1}$ |

Table 1: Medians and upper 95% quantiles for domain generalization L2 errors (i.e. on unseen environments) for different model types and data-generating mechanisms (lower is better).

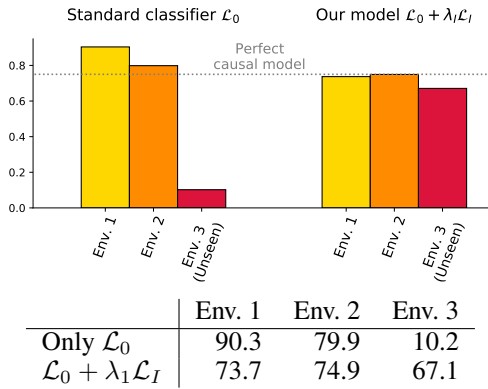

| | Env. 1 | Env. 2 | Env. 3 |
|---|---|---|---|
| Only $\mathcal{L}_0$ | 90.3 | 79.9 | 10.2 |
| $\mathcal{L}_0 + \lambda_1 \mathcal{L}_I$ | 73.7 | 74.9 | 67.1 |

Figure 4: Accuracy of a standard classifier (only $\mathcal{L}_0$, left), and our model ($\mathcal{L}_0 + \lambda_1 \mathcal{L}_I$, right)

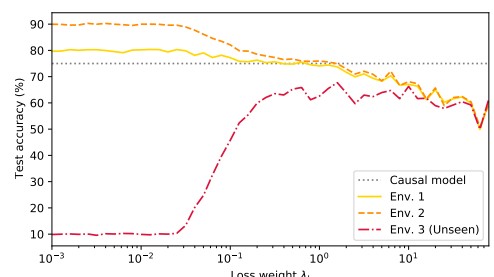

Figure 5: Performance of the model in the three environments, depending on the hyper-parameter $\lambda_1$.

that limits the achievable shape-based classification accuracy to 75%. To confuse the classifier, digits are additionally colored such that colors are spuriously associated with the true labels at accuracies of 90% resp. 80% in the first two environments, whereas the association is only 10% correct in the third environment. A classifier naively trained on the first two environments will identify color as the best predictor, but will perform terribly when tested on the third environment. In contrast, a robust model will ignore the unstable relation between colors and labels and use the invariant relation, namely the one between digit shapes and labels, for prediction. We supplement the HSIC loss with a Wasserstein term to explicitly enforce $R \perp E$, i.e. $\mathcal{L}_I = \text{HSIC} + \text{L2}(\text{sort}(R^{e_1}), \text{sort}(R^{e_2}))$ (see Appendix D). This gives a better training signal as the HSIC alone, since the difference in label-color association between environments 1 and 2 (90% vs. 80%) is deliberately chosen very small to make the task hard to learn. Experimental details can be found in Appendix H. Figure 4 shows the results for our model: Naive training ($\lambda_1 = 0$, i.e. invariance of residuals is not enforced) gives accuracies corresponding to the association between colors and labels and thus completely fails in test environment 3. In contrast, our model performs close to the best possible rate for invariant classifiers in environments 1 and 2 and still achieves 67% in environment 3. The recently proposed REx method from (Krueger et al., 2020) even got 68.7% accuracy on the unseen environment. It will be interesting to investigate whether this is a deeper consequence of their objective or just an insignificant effect of better network tuning and minor differences in experimental protocols. Figure 5 demonstrates the trade-off between goodness of fit in the training environments 1 and 2 and the robustness of the resulting classifier: the model's ability to perform DG to the unseen environment 3 improves as $\lambda_I$ increases. If $\lambda_I$ is too large, it dominates the classification training signal and performance breaks down in all environments. However, the choice of $\lambda_I$ is not critical, as good results are obtained over a wide range of settings.

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
