# OpenReview forum: "Learning Robust Models using the Principle of Independent Causal Mechanisms"
_ICLR.cc/2021/Conference — Reject_

### Official Review · AnonReviewer2 · 2020-10-27
**Well-motivated paper, but more detail needed**

**Rating:** 6
**Confidence:** 4

**Review:**

The paper is well-motivated and studies and important topic, but unfortunately it is let down by the presentation of their contributions which is confusing and at times misleading.

First - a more minor complaint (which I put here because its a source of confusion for the rest of the review) - The normalizing flow section is confusing because the mapping between the base distribution and y isn't clear. I normally think of a normalizing flow as a map from some base distribution u to some target y such that y = T(u) and p(y) = p(u)|det J_T (u) | where u=T^{-1}(y) (adding conditioning as required to make a conditional flow). This paper uses T(Y; h(X)) everywhere - which I think is referring to T^{-1}(Y; h(X)) because we normally think of T as acting on the base distribution U and T^{-1} as acting on the target variable. My review assumes that I should read T(.) as a map from Y -> U... but that's a little weird and should be explained explicitly.

More seriously, I don't understand why Lemma 1 isn't trivial? By the data processing inequality, any transformation of X can only lose information about X. So if the identity function is among the set of feature extractors, then h^* includes it,  because it maximizes I(h(X), Y). The fact that h^* is independent of the flow's latent variable trivially follows from the fact that choosing the identity is always optimal. Of course, things get more complex if there is some constraint on H such that the identity isn't included,  but this isn't discussed. On a second reading, I think that this constraint is meant to come from the Y \perp E | h(X) condition in section 4, but how this condition interacts with Lemma 1 needs to be clearer.

The presentation of the method in section 4 also needs work: the domain generalization problem is presented as the problem of finding h that maximizes the mutual information between Y and h(X) in the worst case environment under the constraint that Y\indep E | h(X). As far as I can tell, the independence constraint is the important part of that objective: under that constraint, it is not clear why I wouldn't want to maximize the average mutual information, or some other objective?

Similarly - it's not clear why theorem 1 is useful until we get to equation (5) (and it took me a couple of reads to realize that this is actually the important step) - on its own, it just essentially says that if we have conditional independence, then applying a 1:1 function maintains that conditional independence.

Having gotten to this point in this review, I think that many of my issues would be resolved if the presentation order was reversed. The key condition you need is Y \perp E | h(X); The paper would be far easier to follow by making it clear that is is the condition you need, explaining both why we can't optimize for it directly, and why this particular normalizing flow approach gives an indirect approach to achieving the condition. In the current order of presentation which leads with a discussion of normalizing flows, we are presented with theoretical results about flows which, in isolation, seem trivial.

The experiments show the method shows promise (though they should report both IRM & REX [Kruger et al 2020]'s performance for coloured MNIST to make it clear that there are better methods on that dataset)...

[Kruger et al 2020] Out-of-Distribution Generalization via Risk Extrapolation

---

> ### Author Response · Authors · 2020-11-17
> **Thanks for the comments (Part 1)**
>
> Thank you for your thoughtful comments. We address each one in turn and will update our draft accordingly.
>
> -> _”The paper is well-motivated and studies an important topic, but unfortunately it is let down by the presentation of their contributions which is confusing and at times misleading.”_
>
> Our main contributions can be summarized as follows:
> - We reformulate the principle of independent causal mechanisms as an objective amenable to continuous optimization methods (section 4.2). Our formulation is more general than Peters et al. (2016) and Rojas-Carulla et al. (2018) because our objective is not restricted to linear models. Moreover, by connecting invariant (causal) relations with normalizing flows (Theorem 1), we transform the objective into a loss that allows for standard gradient-based training.
> - We take advantage of the continuous objective in three important ways: (1) Our $h(x)$ can learn invariant new features, whereas $h(x)$ in a graph-based method can only select features from a pre-defined set. (2) Our approach does not suffer from the scalability problems of combinatorial optimization methods as proposed in e.g. Peters et al. (2016) and Rojas-Carulla et al. (2018). (3) Our optimization via normalizing flows, i.e. in the form of a density estimation task, facilitates accurate maximization of the mutual information in eq. (4).
> - We show how our objective simplifies in important special cases and under which conditions its optimal solution identifies the true causal parents. We empirically demonstrate that the new method achieves good results on two datasets proposed in the literature.
>
> We will make this more clear in an updated version of the paper.
>
>
> -> _“The normalizing flow section is confusing because the mapping between the base distribution and y isn't clear. “_
>
> Due to invertibility, normalizing flows can be used for both generative modeling and density estimation. In a generative model, it is most natural to assign the term “forward direction” to the mapping from the base distribution to the data distribution, as you suggest. However, in density estimation the opposite convention is more natural and therefore more common. Since we learn a density model for $p(y | h(x))$, we opted for the second possibility, i.e. we consider the mapping from data to latent variables as “forward”. The mathematical results are, of course, the same in either case, only the meaning of $T$ and $T^{-1}$ is exchanged.
>
> -> _“More seriously, I don't understand why Lemma 1 isn't trivial?”_
>
> We will rewrite Lemma 1 as follows:
> Let $  h^{\star}, T^{\star}= \arg \min_{h \in \mathcal{H}, T \in \mathcal{T}} \mathcal{L}_{\mathrm{NLL}}  (T, h)$ be the solution of the NLL minimization problem on a sufficiently rich function space $\mathcal{T}$. Then the following properties are guaranteed for arbitrary sets $\mathcal{H}$ of feature extractors:
> - $h^{\star}$ also maximizes the mutual information, i.e. $h^{\star} = g^{\star}$ with $g^{\star}= \arg \max_{g \in \mathcal{H}} I(g(X); Y)$
> -  $h^{\star}$ is independent of the flow's latent variable: $h^{\star}(X) \perp  T^{\star}(Y; h^{\star}(X)) $.
>
> When the identity is an element of $\mathcal{H}$, it is indeed a trivial optimum for $g^{\star}$. However, in our opinion it is not obvious that the maximally informative features $g^{\star}$ (for any permissible function space $\mathcal{H}$) are also the optimal solution $h^{\star}$ of the NLL objective, i.e. that NLL minimization automatically leads to maximally informative features (statement (a)).  The same holds for statement (b): It is not trivial that minimizing the NLL loss results in latent variables/residuals that are independent of the features.

---

> > ### Author Response · Authors · 2020-11-17
> > **Thanks for the comments (Part 2)**
> >
> > -> _“On a second reading, I think that this constraint is meant to come from the $Y \perp E |h(X)$ condition in section 4, but how this condition interacts with Lemma 1 needs to be clearer.”_
> >
> > Indeed, choosing $h(X)$ as the identity is often not permitted because it violates the conditional independence requirement $Y \perp E | h(X)$. It is an important property that the optimal NLL solution $h^{\star}$ under this constraint is still maximally informative. We will make this connection between Lemma 1 and eq. (4) clearer in the paper.
> >
> > -> _“the domain generalization problem is presented as the problem of finding h that maximizes the mutual information between $Y$ and $h(X)$ in the worst case environment under the constraint that Y\indep E | h(X). As far as I can tell, the independence constraint is the important part of that objective: under that constraint, it is not clear why I wouldn't want to maximize the average mutual information, or some other objective?”_
> >
> > Our method could indeed maximize the average mutual information as well. However, under this objective, very good performance in a majority of the environments could mask failure in a single (outlier) environment. We therefore opted for optimizing w.r.t. the worst environment (which can change in the course of optimization).
> >
> > -> _“Similarly - it's not clear why theorem 1 is useful until we get to equation (5) (and it took me a couple of reads to realize that this is actually the important step) - on its own, it just essentially says that if we have conditional independence, then applying a 1:1 function maintains that conditional independence.”_
> >
> > To clarify the importance of Theorem 1, we propose to insert the following statement just above:
> > “The following theorem shows that suitable normalizing flows will translate statistical independence of the latent variable/residual $R$ from the environment $E$ and the features $h(X)$ into the desired invariance condition of the mechanism $p(Y | h(X))$ from the environment $E$.”
> >
> > -> _“Having gotten to this point in this review, I think that many of my issues would be resolved if the presentation order was reversed.“_
> >
> > We suggest presenting the DG objective (eq. 4) before Lemma 1, so that the motivation behind this lemma, as well as its connection to the invariance property, become clear. Would this resolve your confusion?
> >
> > -> _“The experiments show the method shows promise (though they should report both IRM & REX [Kruger et al 2020]'s performance for coloured MNIST to make it clear that there are better methods on that dataset).”_
> >
> > Thanks for pointing out this interesting reference. We will add the citation and mention that REX works better.
> >
> > References:
> > - Mateo Rojas-Carulla, Bernhard Schölkopf, Richard Turner, and Jonas Peters.  Invariant models for causal transfer learning.The Journal of Machine Learning Research, 19(1):1309–1342, 2018.
> > - Jonas Peters, Peter Bühlmann, and Nicolai Meinshausen. Causal inference by using invariant prediction: identification and confidence intervals. Journal of the Royal Statistical Society: Series B (Statistical Methodology), 78(5):947–1012, 2016.

---

> > > ### Comment · AnonReviewer2 · 2020-11-24
> > > **Response**
> > >
> > > Sorry about the delay in responding. Thanks for the clarifications. I think with the modifications that you propose, this paper is worth publishing and will update my score to a 6 in response.

---

### Official Review · AnonReviewer4 · 2020-10-30
**The assumptions are very strong, yet a good implementation method is proposed**

**Rating:** 6
**Confidence:** 4

**Review:**

In this paper, the authors propose a gradient-based learning framework, with a two part objective function in which one part improves the informativeness about the target variable, and the other part enforces the invariance of the relation. The second part is based on the ICM principle and increases the stability and renders domain generalization possible.

The paper is well written and, for the most part, is easy to follow.

We should note that the ICM principle is only usable if we have no hidden confounders, i.e., causal sufficiency, in the system. The authors should clarify that causal sufficiency is an important assumption early in the manuscript and should clarify what will happen to the results if it is violated.

In general, the assumptions in this work are very strong and I do not believe they will hold in reality. Specially, regarding Assumption 2, if we are assuming some of the causal mechanisms are changing across environments, why the one corresponding to the target should not change?

Although the assumptions are strong, same assumptions were considered in few other works such as (Peters et al., 2016). Compares to existing work with the same assumptions, this paper provides a good implementation method that is an improvement over past work and would be of interest to the ICLR community.

The authors also discuss the conditions under which the recovered stable relations correspond to the true causal mechanisms. The use of ICM for causal discovery is also extensively studied in the non-parametric case in [Huang et al., Causal Discovery from Heterogeneous/Nonstationary Data], and in the linear case in [Ghassami et al., Multi-domain Causal Structure Learning in Linear Systems].

The definition of do-intervention in page 3 is not standard. What is referred to as do intervention in this paper is usually referred to as hard intervention in the literature, and what is referred to as hard intervention in this paper is usually referred to as atomic intervention in the literature.

---

> ### Author Response · Authors · 2020-11-17
> **Thanks for the comments**
>
> Thank you for your thoughtful comments. We address each one in turn and will update our draft accordingly.
>
> -> _“In general, the assumptions in this work are very strong and I do not believe they will hold in reality. Specially, regarding Assumption 2, if we are assuming some of the causal mechanisms are changing across environments, why the one corresponding to the target should not change?”_
>
> We agree that the assumptions are quite strong, but believe that domain generalization is in general impossible without strong assumptions (in contrast to classical supervised learning). In our view, the interesting question is “Which strong assumptions are the most useful in a given setting?” For example, Ghassami et al. (2017) use a more relaxed notion of invariance, but need to compensate for this by imposing linear causal mechanisms.
>
> Under this premise, we are not so pessimistic about the realism of our assumptions. For instance, Heinze-Deml et al. (2018) use assumption 2 to identify causes for birth rates in different countries. If all variables mediating the influence of continent/country (environment variable) on birth rates (target variable) are included in the model (e.g. GDP, Education…), this assumption is reasonable. The same may hold for epidemiological investigations. Pfister et al. (2019) suppose assumption 2 in the field of finance.
>
> Another reasonable example are data augmentations in computer vision. Deliberate image rotations, shifts and distortions can be considered as environment interventions that preserve the relation between semantic image features and object classes (see e.g. Mitrovic et al. 2020), i.e. verify assumption 2. In general, assumption 2 may be justified when one studies a fundamental mechanism that can reasonably be assumed to remain invariant across environments, but is obscured by unstable accidental relationships between observable variables.
>
> -> _“We should note that the ICM principle is only usable if we have no hidden confounders, i.e., causal sufficiency, in the system. The authors should clarify that causal sufficiency is an important assumption early in the manuscript and should clarify what will happen to the results if it is violated.”_
>
> Thanks for bringing up this point. Causal sufficiency is indeed an important assumption and we will clarify this in the paper. For instance, a causal graph of the form $H\rightarrow X_1$, $X_1\rightarrow Y$ and $H\rightarrow Y$, where $H$ is not observed, violates the causal sufficiency assumption. If the environment influences $X_1$ (i.e. the graph also contains edge $E\rightarrow X_1$) and the generated distribution satisfies the Causal Markov Condition, it follows that $E \perp Y | X_1$  is unachievable.
> Huang et al. (2020) allows for restricted confounding, but our method is not yet capable of doing the same.
>
> -> _“The authors also discuss the conditions under which the recovered stable relations correspond to the true causal mechanisms. The use of ICM for causal discovery is also extensively studied in the non-parametric case in [Huang et al., Causal Discovery from Heterogeneous/Nonstationary Data], and in the linear case in [Ghassami et al., Multi-domain Causal Structure Learning in Linear Systems].”_
>
> Thank you for pointing out these references. Huang et al. (2020), among other qualities, is remarkable for its robustness against non-stationary behavior and a limited amount of hidden confounding, which our method can not (yet) deal with. On the other hand, as a graph-based method it requires all variables to be pre-defined (as nodes of the graph) and suffers from the limited scalability of combinatorial optimization procedures, whereas we can learn new features through our function $h(x)$ and rely on gradient-based optimization with much better scaling properties for large problems.
> Ghassami et al. (2018) use a different form of invariance assumption which relies strongly on the linearity assumption. We will add these references in the paper.
>
> -> _“The definition of do-intervention in page 3 is not standard.”_
>
> This will be corrected in the updated version.
>
> References:
> - C. Heinze-Deml,  J. Peters,  and  N. Meinshausen.   Invariant  causal  prediction  for nonlinear models.Journal of Causal Inference, 6(2), 2018.
> - J. Mitrovic, B. McWilliams, J.Walker, L. Buesing, C. Blundell. Representation Learning via Invariant Causal Mechanisms. arXiv:2010.07922, 2020
> - A. Ghassami, S. Salehkaleybar, N. Kiyavash, and K. Zhang. Learning causal structures using regression invariance. InAdvances in Neural Information Processing Systems, pp.3011–3021, 2017.
> - N. Pfister, P. Bühlmann, and J. Peters. Invariant causal prediction for sequential data. Journal of the American Statistical Association, 114(527):1264–1276, 2019.
> A. Ghassami, N. Kiyavash, B. Huang and K. Zhang. Multi-domain Causal Structure Learning in Linear Systems. InAdvances in Neural Information Processing Systems,  2018.

---

### Official Review · AnonReviewer3 · 2020-10-31
**An effective exposition of key ideas in causal inference, but novelty should be clearer**

**Rating:** 6
**Confidence:** 4

**Review:**

The paper presents a new gradient-based framework for learning invariant mechanisms (often called "relations" in the paper) from data drawn for multiple environments (data generating processes).  Overall, the writing is excellent, and the central ideas are interesting and valuable.

A key idea of the paper is that training data drawn from different environments can be exploited to learn mechanisms that remain invariant across those environments.  While true, this is unsurprising and well-established.  Fundamental principles of causal inference, known for decades at this point, directly imply that different environments (data generating processes with different interventions) will allow identification of different sets of causal dependencies.  Practical methods for such identification have been demonstrated using graphical models and relatively simple methods for parameterization of those models.  The paper could be improved by spending less time on the known results (or at least making clearer connections to prior work) and spending more time clarifying what is genuinely novel about the proposed ideas.  In addition, the authors should make a greater effort to distinguish between central ideas and implementation details.

Multiple times in the paper, basic results from the causal inference literature are attributed to relatively recent papers (e.g., Peters et al. (2017)), including the special properties of the causal factorization and the idea of invariance of mechanisms in response to intervention.  These ideas can be traced back much further.  For example, the basic idea of invariance to intervention (so called "autonomy" or "modularity") has been known since at least the 1930s.  Heckman and Pinto (2015) note that: "In the language of Frisch (1938), these structural equations are autonomous mechanisms represented by deterministic functions mapping inputs to outputs. By autonomy we mean, as did Frisch, that these relationships remain invariant under external manipulations of their arguments." The paper would be improved by making clearer when concepts were first identified and by who.

The empirical evidence provided for the claims in the paper is relatively modest.  The simulated results provided in Table 1 shows only very small differences in L2 errors among variants of the authors' proposed methods, and more substantial improvements over ICP and ERM (in three of four cases).  The discussion of these results is excellent.  The results on the "Colored MNIST" data show the expected results.  However, good performance on simulated data and only a single real data set is still relatively weak evidence for the claims made in the paper.  The paper would be improved by increasing the number of real data sets used for evaluation.

References

Heckman, J., & Pinto, R. (2015). Causal analysis after Haavelmo. Econometric Theory, 31(1):115-151.

---

> ### Author Response · Authors · 2020-11-17
> **Thanks for the comments**
>
> Thank you for your thoughtful comments. We address each one in turn and will update our draft accordingly.
>
> -> _"The paper could be improved by spending more time clarifying what is genuinely novel about the proposed ideas. In addition, the authors should make a greater effort to distinguish between central ideas and implementation details."_
>
> Our main contributions can be summarized as follows:
> - We reformulate the principle of independent causal mechanisms as an objective amenable to continuous optimization methods (section 4.2). Our formulation is more general than Peters et al. (2016) and Rojas-Carulla et al. (2018) because our objective is not restricted to linear models. Moreover, by connecting invariant (causal) relations with normalizing flows (Theorem 1), we transform the objective into a loss that allows for standard gradient-based training.
> - We take advantage of the continuous objective in three important ways: (1) Our $h(x)$ can learn invariant new features, whereas $h(x)$ in a graph-based method can only select features from a pre-defined set. (2) Our approach does not suffer from the scalability problems of combinatorial optimization methods as proposed in e.g. Peters et al. (2016) and Rojas-Carulla et al. (2018). (3) Our optimization via normalizing flows, i.e. in the form of a density estimation task, facilitates accurate maximization of the mutual information in eq. (4).
> - We show how our objective simplifies in important special cases and under which conditions its optimal solution identifies the true causal parents. We empirically demonstrate that the new method achieves good results on two datasets proposed in the literature.
> We will make these points more clear and differentiate them from implementation details in an updated version of the paper.
>
> -> _"Practical methods for such identification have been demonstrated using graphical models and relatively simple methods for parameterization of those models."_
>
> Graphical models are, of course, the classical approach to causal analysis. We consider our method as an interesting alternative for three reasons: (1) Graphical models have to consider the system as a whole, whereas we can focus on a single causal mechanism of interest. (2) Graphical models need pre-defined features (=nodes), whereas we can learn new invariant features via the function $h(X)$. (3) Graphical models rely on combinatorial optimization methods, which often scale poorly to large problems, whereas our continuous optimization suffers much less from this shortcoming.
>
> -> _“Multiple times in the paper, basic results from the causal inference literature are attributed to relatively recent papers (e.g., Peters et al. (2017)), including the special properties of the causal factorization and the idea of invariance of mechanisms in response to intervention. These ideas can be traced back much further.”_
>
> We often cite Peters et al. (2017) as a textbook, since it provides a concise collection and review of relevant prior work. We opted for this citation style mainly for the convenience of the reader and do not mean to imply that all their material is novel. We will clarify in the paper that many ideas go much further back.
> Moreover, while the idea of causal relations to remain invariant across environments has been known for a long time, we would like to emphasize that its converse -- using invariance to detect causal relations -- was only exploited recently. In the words of Bühlmann (Invariance, Causality and Robustness, 2018 Neyman Lecture): “The reverse relation ‘causal structures ← invariance’ has not been considered until recently (Peters et al., 2016)”.
>
> -> _“The paper could be improved by spending less time on the known results.”_
>
> We included some known results in order to make the paper more accessible to the non-specialist in causal modeling. We will try to streamline these sections in an updated version of our paper.
>
> -> _"The paper would be improved by increasing the number of real data sets used for evaluation."_
>
> We fully agree, but the requirements for a convincing real-world test dataset are non-trivial. Although we think that many interesting real-world applications exist, no causal ground truth is available for them (e.g. the datasets in Huang et al. (2020)), limiting their value for systematic evaluation of our approach. Do you have suggestions for suitable datasets?
>
> References:
> - Bühlmann. Invariance, Causality and Robustness, 2018 Neyman Lecture
> - Mateo Rojas-Carulla, Bernhard Schölkopf, Richard Turner, and Jonas Peters.  Invariant models for causal transfer learning.The Journal of Machine Learning Research, 19(1):1309–1342, 2018.
> - Huang, Biwei, et al. "Causal discovery from heterogeneous/nonstationary data." Journal of Machine Learning Research 21.89 (2020): 1-53.

---

### Author Response · Authors · 2020-11-24
**Update Summary**

We thank the reviewers again for their thoughtful comments and have revised the manuscript accordingly. Text in blue highlights our additions and changes.
The main updates for the submission are the following:
- Elaborate on the contributions and novelty of our approach. (Section 1, R2 and R3)
- Add further references to the related work and make clear that some ideas go much further back. (Section 2, R3)
- Use standard terminology for interventions. (Section 3.1, R4)
- Discuss the role of hidden confounders for assumption 2 and their consequences for our approach. (Section 3.1 and Section 4.1, R4)
- Restructure the theoretical derivation by introducing the DG objective at the beginning of Section 3.2, before Lemma 1 and Theorem 1. This clarifies how our theoretical results overcome the difficulties in the original DG formulation. (Section 3, R2)
- Mention the possibility to optimize for other objectives (e.g. average mutual information rather than maximin). (Section 3.2, R2)
- Improve the introduction of normalizing flows and their relationship to structural equations. Specifically, we clarify the distinction between the generative direction (from base to data distribution) and the density estimation direction (from data to base distribution). (Section 3.3, R2)
- Re-phrase Lemma 1 to make clear that it is non-trivial and improve the discussion of its impact on reformulating the DG objective. (Section 3.3, R2)
- Revise the motivation of Theorem 1 according to your suggestions. (Section 4.1, R2)
- Add and discuss REx in the experimental part. (Section 5.2, R2)

---

### Decision · Program_Chairs · 2021-01-07
**Final Decision**

**Decision:**

Reject

**Comment:**

This paper proposes a new framework for improving supervised learning via invariant mechanisms. The reviewers agree that overall, this paper is well-written and contributes to a growing body of work on invariant prediction and causality in supervised learning. At the same time, there are some concerns regarding novelty and significance in light of previous work, as well as the overall organization of the paper, which could be improved to highlight the main contributions more clearly. Ultimately, this was a borderline decision, but it is clear that the paper needs a major revision before acceptance. Although the authors have already incorporated some of the minor comments which is appreciated, the authors are urged to consider the major comments (e.g. see R2's comments regarding presentation) when revising the paper.